# Inflammatory bowel disease and risk of idiopathic pulmonary fibrosis: A protocol for systematic review and meta-analysis

Jiali Wang[1,2☯], Fushun Kou[1,2☯], Xiao Han[1,2], Lei Shi[2], Rui Shi[2], Zhibin Wang[2], Tangyou Mao[2], Junxiang Li[2]*

1 Graduate School, Beijing University of Chinese Medicine, Beijing, China, 2 Department of Gastroenterology, Dongfang Hospital, Beijing University of Chinese Medicine, Beijing, China

☯ These authors contributed equally to this work.

* lijunxiang1226@163.com

## Abstract

### Introduction

Inflammatory bowel disease is a relapsing chronic gastrointestinal inflammatory disease. Idiopathic pulmonary fibrosis is a rare but serious extraintestinal pulmonary manifestation of inflammatory bowel disease. However, the relationship between these two conditions is unclear. Therefore, this study aims to elucidate this relationship through a systematic review and meta-analysis, focusing on the risk of idiopathic pulmonary fibrosis in patients with inflammatory bowel disease.

### Methods

The systematic review will be outlined according to the Preferred Reporting Items for Systematic Review and Meta-Analyses Protocols and its extension statement for reporting systematic reviews incorporating network meta-analyses of healthcare interventions: checklist and explanations. Original articles published in any language will be searched in the following databases: PubMed, Web of Science, EMBASE, Google Scholar, and Ovid. Observational studies that reveal an association measure between idiopathic pulmonary fibrosis and inflammatory bowel disease will be included (cross sectional, cohort, and case-control trials). Two independent reviewers will be assigned to evaluate study quality using the Newcastle–Ottawa scale for assessing the quality of non-randomized studies in meta-analyses. Sensitivity analyses will be conducted based on the quality of included studies. All relevant studies will be assessed based on the study type, sample size, inflammatory bowel disease subtype, odds ratio, confidence interval, treatment strategy, and follow-up. The Grading of Recommendations Assessment, Development, and Evaluation approach will be used to rate the quality of the evidence.

### Discussion

The results of this meta-analysis may show that patients with inflammatory bowel disease are at higher risk of developing idiopathic pulmonary fibrosis. This study will be the first

**Editor:** Negar Rezaei, Non-Communicable Diseases Research Center, Endocrinology and Metabolism Population Sciences Institute, Tehran University of Medical Sciences, ISLAMIC REPUBLIC OF IRAN

**Data Availability Statement:** No datasets were generated or analysed during the current study. All

relevant data from this study will be made available upon study completion.

**Funding:** The study was supported by a grant from the the National Key R&D Program of China (2018YFC1705403).

**Competing interests:** The authors have declared that no competing interests exist.

meta-analysis to focus on the association between inflammatory bowel disease and idiopathic pulmonary fibrosis. Exploring the relationship between the two conditions may further enhance our understanding of the pathogenesis of inflammatory bowel disease and idiopathic pulmonary fibrosis and promote the development of related research fields.

## Introduction

Inflammatory bowel disease (IBD) is a relapsing chronic inflammatory disease mediated by immune responses. There are two types—ulcerative colitis (UC) and Crohn's disease (CD) [1]. The pathogenesis of IBD is not fully understood, but it is assumed to involve the intestinal flora, intestinal mucosal permeability, immune disorders, environment, and genetics [1, 2]. An increasing number of epidemiological studies have indicated that IBD is a health and economic burden for people worldwide [3]. Moreover, although the incidence of IBD has increased and then plateaued in the Western world during the past 100 years, its prevalence has risen rapidly in the East, with more patients presenting with complicated disease [4–6].

Extraintestinal manifestations of IBD in the lungs include interstitial lung disease, granulomatous lung disease, and eosinophilic pneumonia [7, 8]. In potentially related pulmonary diseases, idiopathic pulmonary fibrosis (IPF) is a chronic, progressive, and fibrotic interstitial lung disease of unknown etiology that ultimately leads to death from respiratory failure. It tends to occur more commonly in older people and has a high mortality rate [9, 10]. Existing research data indicate that the prevalence and incidence of IPF have increased globally, particularly with age and especially in men compared to that in women [11, 12]. Recently, a large national epidemiological survey in South Korea has showed that the risk of developing IPF in patients with IBD is rising, and it is more obvious in patients with CD [13]. Although most small cohort studies have limited the depth of IPF research [14], more and more IBD lung manifestations have been reported, which deserve our attention [15–17].

According to existing literature, the risk and magnitude of IPF in patients with IBD are unclear. More research has been conducted; however, no systematic review has been conducted to elucidate this. Therefore, a systematic review and meta-analysis on this theme is necessary.

In recent decades, meta-analyses have been widely used to summarize literature, reach sound conclusions, and guide the establishment of clinical policies and guidelines. We propose this meta-analysis protocol as a feasible approach to explore the risk of IPF in patients with IBD. The results obtained will shed light on how these conditions relate to each other, aiding to identify the risk factors, prevention strategies, and treatments.

## Methods

### Study design and registration

This systematic review and meta-analysis aims to explore the risk of IPF in patients with IBD. The protocol has been developed according to the Preferred Reporting Items for Systematic Review and Meta-Analyses Protocols (PRISMA-P) [18] and the PRISMA extension statement for reporting systematic reviews incorporating network meta-analyses of healthcare interventions: checklist and explanations [19]. According to PRISMA-P, the study has been registered with the International Prospective Register of Systematic Reviews Network (PROSPERO, CRD 42020169014).

## Search strategy

Two independent researchers will search the PubMed, Web of Science, EMBASE, Google Scholar, and Ovid databases for original articles without language restrictions published as of January 10, 2022. The search terms will include inflammatory bowel disease, IBD, idiopathic pulmonary fibrosis, and IPF. The search strategy is IPF (or related terms, MeSH major topic) AND IBD (or related terms, MeSH major topic). The detailed search strategy for PubMed is presented in S1 Table. The search terms will be appropriately adjusted to fit the grammar rules of different databases. Additional resources including lists of references in all major research and review articles on the topic, erratum, and retraction rates of the included studies published in PubMed will be searched and the completion date of the above will be recorded in the review.

## Inclusion criteria

Observational studies, including cross-sectional, cohort (retrospective or prospective), and case-control studies that reveal association measures between IBD and IPF and include adult participants (aged >18 years) with IPF (defined as pulmonary fibrosis without identifiable etiology) will be searched. Studies involving multiple lung lesions will be included, but only those with IPF data will be analyzed. IBD must have been diagnosed on a clinical basis or by colonoscopy. The exposure factor will be IBD. The control group will comprise either the general population or participants without IBD. The studies will not include restrictions in sample size, follow-up period, article language or publication status. The main outcome of our study will be to evaluate the risk of IPF in patients with IBD.

## Exclusion criteria

Studies in any of the following categories will be excluded: experimental animal studies, review articles, case reports, studies without control group, studies with incomplete data, or editorials and conference abstracts. Additionally, in case of multiple studies of the same sample, the study with the largest sample size and the longest follow-up period will be analyzed.

## Study selection

The data retrieved from each database will be imported into EndNote X9 software, and will be checked and deduplicated. Titles and abstracts of deduplicated articles will be read by two independent reviewers and initially screened based on inclusion criteria. If a study meets the inclusion criteria after the preliminary screening, the full text will be individually rechecked by two reviewers according to the predetermined criteria and finally included or excluded. Unverifiable literature and disagreements will be discussed between the two reviewers. If the two cannot reach a consensus, a third reviewer will step in. The author will be contacted if there is no full text or more details are needed. The process will follow the PRISMA flow diagram [20].

## Data extraction

A table will be designed using Excel 2016 to extract information from the included studies. Data will be extracted separately by two independent reviewers. The data extraction table will include information on author name, contact information, journal name, year of publication, study characteristics (study design, sample size, study period, follow-up period, and country where study was conducted), demographic characteristics (age, sex, disease stage, and severity), study exposure, and outcome characteristics.

Subsequently, all relevant studies will be compared according to the study type, sample size, subtype of IBD, odds ratio (OR), 95% confidence interval (CI), treatment strategy, and follow-up. In case of a lack of any relevant data, the author(s) of the concerned study will be contacted directly.

## Assessing risk of bias

The Newcastle–Ottawa scale will be used to assess the bias of non-randomized controlled trials in the meta-analysis [21]. The assessment of bias in the case-control study will include the selection of case and control groups, comparability, and exposure. The bias assessment in cohort studies will include cohort selection, comparability, and outcome. 'Low', 'medium' and 'high' quality studies will be represented with a score of 0–3, 4–6 and 7–9, respectively. Cross-sectional studies were evaluated according to the checklist advised by the Agency for Healthcare Research and Quality (https://www.ncbi.nlm.nih.gov/books/NBK35156/). The quality assessment is divided into three levels: 'low', 'medium' and 'high', and the scores correspond to 0–3, 4–7 , and 8–11 in turn. Bias assessments will be performed by two researchers; if the two hold different opinions, the issue will be settled by discussion until an agreement is reached.

## Strategy for data synthesis

The currently available literature, including data on author name, year of publication, type of study, design, results of interest, sample size, author's conclusions, and main findings will be reported. Qualitative evidence synthesis will be performed based on the available results. After describing the baseline characteristics of the studies, the outcome of interest will be summarized, that is, the association between IBD and the risk of IPF. Furthermore, the associations between CD and UC and the risk of IPF will be assessed separately.

The Cochrane Review Manager software RevMan and the statistical software SPSS-25 will be used to assess the data. If quantitative research is not appropriate, qualitative research will be adopted to describe the included research. Forest plots will be created to intuitively evaluate the effect size and corresponding 95% CI. Weighted mean differences with 95% CI will be used for continuous outcomes; OR or RR with 95% CI will be used for dichotomous outcomes.

Heterogeneity will be assessed based on the results of the $Chi^2$ test and the I2 statistic. Homogeneity will be defined if the $Chi^2$ test of P-value was >0.1 and the $I^2$ value is <50%, and the fixed effects model will be adopted; otherwise, the random effects model will be adopted to integrate the meta-analysis data.

## Subgroup analysis

Subgroup analyses will be performed to analyse the causes of heterogeneity. Subgroup analysis should include but not be limited to study type, age, sex, country, ethnicity, sample size, and follow-up period. Additionally, a potential source of heterogeneity should be considered, which is the method to achieve a diagnosis of IBD and IPF (medical records/ICD criteria/prospective studies with histology/multidisciplinary evaluation). Furthermore, other subgroup analyses will be considered in the study, including tobacco exposure, a potential analysis by cofounder for IBD and IPF.

## Sensitivity analysis

Sensitivity analyses will be proceeded based on the quality of the included studies to determine potential reasons of heterogeneity by omitting studies one by one and evaluating the resulting effect.

## Assessment of publication bias

Publication bias will be assessed using funnel plots and Egger tests if >10 studies are included. If the funnel plot shows asymmetry, it will indicate publication bias. If publication bias exists, trim and fill analyses will be used to assess the impact of publication bias on the results. Any bias will be explained through the analyses and discussions.

## Quality of evidence

The Grading of Recommendations Assessment, Development, and Evaluation (GRADE) approach will be used to assess the quality of the evidence [22]. The GRADE includes five factors that can reduce the quality of the evidence—limitations in designing the study or its execution (risk of bias), inconsistent results, indirectness of evidence, lack of precision, and publication bias—and three factors that can increase the quality of the evidence—very obvious therapeutic effect, conclusive evidence for a dose-response relationship, or all reasonable biases reduce the true treatment effect. Two independent researchers will rate the quality of the evidence for each result as high, moderate, low, or very low according to GRADE.

## Patient and public involvement

This research will be based solely on published research, so patients and the public will not be directly involved.

## Ethics and dissemination

This study will not be related directly to any patients or raise ethical concerns and, therefore, does not require ethics committee approval. The results will be published and disseminated in a peer-reviewed journal article.

## Discussion

This study aims to assess the risk of IPF in patients with IBD. The results of this meta-analysis may show an association between IBD and IPF and suggest that patients with IBD are at a higher risk of developing IPF.

In recent years, the incidence of IBD and IPF has increased worldwide. Previous studies have not clearly articulated the pathogenesis of these two factors or their correlation. However, with increasing research, evidence has shown that the two may be linked. Some factors that contribute to the extraintestinal manifestations of IBD may cause IPF [23, 24]. Both IBD and IPF are associated with abnormal coagulation and are prone to thromboembolic disease. With the development of the disease, both are prone to malignant transformation to form tumors. Some studies have found that excessive Toll-interacting protein expression may contribute to the deterioration of IPF, whereas decreased expression may be associated with the occurrence of IBD [25]. Interleukin-25 and CDKN2B-AS1 are associated with IPF and IBD [26, 27]. In terms of treatment, studies have found that drugs used to treat IPF, such as pirfenidone, may be useful for intestinal fibrosis, a complication of CD [28–31]. Furthermore, calemin, integrin, and G protein-coupled receptor 84 may be targets for the treatment of both [30, 32, 33]. Fat stem/stromal cell and stromal vascular fraction treatment can improve IBD and IBF by reducing inflammation and promoting tissue repair [34]. However, existing literature is insufficient to account for the relationship between the two. Therefore, further exploration of this relationship is necessary.

To our knowledge, this study will be the first systematic review and meta-analysis focusing on the association between IBD and IPF. Strict adherence to the PRISMA-P guidelines and the

use of a combination of quantitative and qualitative analyses are strengths of this study. A limitation is that there is currently limited information describing a relationship between IPF in IBD, and therefore, the studies may not be sufficient to indicate whether there is a true association between the two. Regardless of the outcome, exploring the relationship between the two may further enhance our understanding of IBD pathogenesis. If our findings indicate an association, they may provide new strategies for the diagnosis, treatment, and prevention of IBD and may promote the development of related research fields. We will issue a statement regarding whether there are important amendments to this protocol in the future.

## Supporting information

**S1 File. PRISMA-P checklist.**
(DOCX)

**S1 Table. Search strategy for PubMed.**
(DOCX)

## Acknowledgments

Thanks to Yuan Cheng, Hui Jiang, and Yi Wang for their suggestions on methods.

## Author Contributions

**Conceptualization:** Jiali Wang, Fushun Kou.

**Formal analysis:** Jiali Wang, Xiao Han.

**Funding acquisition:** Junxiang Li.

**Investigation:** Jiali Wang, Fushun Kou.

**Methodology:** Jiali Wang, Fushun Kou.

**Project administration:** Lei Shi.

**Supervision:** Rui Shi, Zhibin Wang, Tangyou Mao.

**Writing – original draft:** Jiali Wang.

**Writing – review & editing:** Jiali Wang, Lei Shi.

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
