## [Decision Letter · Decision Letter 0]

12 May 2022

PONE-D-22-02613Inflammatory bowel disease and risk of idiopathic pulmonary fibrosis, a protocol for  systematic review and meta-analysisPLOS ONE

Dear Dr. Li,

Thank you for submitting your manuscript to PLOS ONE. After careful consideration, we feel that it has merit but does not fully meet PLOS ONE’s publication criteria as it currently stands. Therefore, we invite you to submit a revised version of the manuscript that addresses the points raised during the review process. Please ensure that your manuscript justified on PLOS ONE’s publication criteria .

We look forward to receiving your revised manuscript.

Kind regards,

Negar Rezaei, M.D., Ph.D.,

Academic Editor

PLOS ONE

Journal Requirements:

4. Thank you for submitting the above manuscript to PLOS ONE. During our internal evaluation of the manuscript, we found significant text overlap between your submission and the following previously published works.

- https://www.worldgastroenterology.org/guidelines/inflammatory-bowel-disease-ibd/inflammatory-bowel-disease-ibd-english

- https://openres.ersjournals.com/content/6/4/00391-2020.full

- https://link.springer.com/article/10.1007/s11739-019-02195-0

- https://www.atsjournals.org/doi/full/10.1164/rccm.201807-1255ST

- https://journals.lww.com/md-journal/Fulltext/2020/08140/Association_between_hyperuricemia,_gout,_urate.50.as

Please revise the manuscript to rephrase the duplicated text, cite your sources, and provide details as to how the current manuscript advances on previous work. Please note that further consideration is dependent on the submission of a manuscript that addresses these concerns about the overlap in text with published work.

Reviewers' comments:

Reviewer's Responses to Questions

**Comments to the Author**

1. Does the manuscript provide a valid rationale for the proposed study, with clearly identified and justified research questions?

Reviewer #1: Partly

Reviewer #2: Yes

Reviewer #3: Yes

2. Is the protocol technically sound and planned in a manner that will lead to a meaningful outcome and allow testing the stated hypotheses?

Reviewer #1: Yes

Reviewer #2: Yes

Reviewer #3: Yes

3. Is the methodology feasible and described in sufficient detail to allow the work to be replicable?

Reviewer #1: Yes

Reviewer #2: Yes

Reviewer #3: Yes

4. Have the authors described where all data underlying the findings will be made available when the study is complete?

Reviewer #1: Yes

Reviewer #2: Yes

Reviewer #3: Yes

5. Is the manuscript presented in an intelligible fashion and written in standard English?

Reviewer #1: Yes

Reviewer #2: No

Reviewer #3: Yes

6. Review Comments to the Author

You may also provide optional suggestions and comments to authors that they might find helpful in planning their study.

Reviewer #1: The authors described a protocol for systematic review and meta-analysis for

IBD and the risk of idiopathic pulmonary fibrosis. I suggest, the authors consider the following points.

1) The introduction section (paragraphs two and three) has some grammatical problems. Language editing is required.

2) Trim and fill analysis would be helpful in case of existing publication bias.

Reviewer #2: Inflammatory bowel disease and risk of idiopathic pulmonary fibrosis, a protocol for

systematic review and meta-analysis

This article presents the study protocol of a systematic review and meta-analysis aims to find the association between inflammatory bowel disease (IBD) and idiopathic pulmonary fibrosis (IPF) risk.

The subject is novel and interesting, but the authors must correct and explain these points:

General comment

1. The manuscript text suffers from fragmentary sentences and poor cohesion. The quality of scientific writing needs attention. Please consider a major revision of the manuscript by a native English speaker with a demonstrated history of scientific writing.

Abstract

1. Introduction: The rationale of the study is not well presented. Please revise.

2. Methods: Please consider using passive voice when presenting methods.

3. Results: Need to be re-written.

Strengths and limitations of this study

1. Needs to be re-written.

Methods

1. Please include relevant references, where necessary.

Discussion

1. Please review the discussion, check its storyline, and improve its coherency. It is not easy to follow in its current form.

2. Please include relevant references, where necessary.

Reviewer #3: Thanks for sending me this interesting SRMA protocol. I think the quality of this paper is suitable for publication. Here are some suggestions:

• Abbreviations should be in complete form at their first appearance

• The English wording and the narrative synthesis should be improved.

• The flow of narrative, how the study was conducted, and the overall structure and sectioning of methods and other parts are acceptable.

• I have no more suggestions to disclose and would be happy to do re-reviewing of the modifications.

7. PLOS authors have the option to publish the peer review history of their article (what does this mean?). If published, this will include your full peer review and any attached files.

Reviewer #1: No

Reviewer #2: **Yes: **Mohsen Abbasi-Kangevari

Reviewer #3: **Yes: **Esmaeil Mohammadi

---

## [Author Response · Author response to Decision Letter 0]

1 Jun 2022

Dear editors:

Thank you for your comments concerning our manuscript. Those comments are all valuable and very helpful for revising and improving our paper, as well as the important guiding significance to our researches. We have done the full English language editing. Specific modifications can be quickly found based on the line number in new "Manuscript". For the full text of the revision changes, see the "Revised Manuscript with Track Changes" file. We hope that the revised new manuscript will be approved.

1.Response: Funding information has not changed, so there is no need to update the statement in the cover letter.

2.Guidelines for resubmitting your figure files are available below the reviewer comments at the end of this letter.

Response: The figure file was deleted. The PRISMA flow diagram takes the form of references.

3.Please ensure that your manuscript meets PLOS ONE's style requirements, including those for file naming. The PLOS ONE style templates can be found at 

Response: The full text has been revised in accordance with the journal format requirements.

4.Your ethics statement should only appear in the Methods section of your manuscript. If your ethics statement is written in any section besides the Methods, please delete it from any other section. 

Response: The ethical statement has been removed from sections besides the Methods. 

5.Please include captions for your Supporting Information files at the end of your manuscript, and update any in-text citations to match accordingly. Please see our Supporting Information guidelines for more information: http://journals.plos.org/plosone/s/supporting-information. 

Response: The table "Search strategy of PubMed" in the original manuscript has been modified as supplementary information. According to the requirements of the journal, the table is named "S1_Table.docx". This manuscript is a protocol for systematic review and meta-analysis, and the PRISMA-P checklist should be used. We have added this checklist as supporting information as required. Because the PRISMA-P checklist requires authors contributions and fundings information, we add them before the references in manuscript. The PRISMA-P checklist is named "S2_File.docx", as requested. The title of Supporting Information files are appended to the end of the manuscript.

6.Thank you for submitting the above manuscript to PLOS ONE. During our internal evaluation of the manuscript, we found significant text overlap between your submission and the following previously published works.

-https://www.worldgastroenterology.org/guidelines/inflammatory-bowel-disease-ibd/inflammatory-bowel-disease-ibd-english

- https://openres.ersjournals.com/content/6/4/00391-2020.full

- https://link.springer.com/article/10.1007/s11739-019-02195-0

- https://www.atsjournals.org/doi/full/10.1164/rccm.201807-1255ST

-https://journals.lww.com/md-journal/Fulltext/2020/08140/Association_between_hyperuricemia,_gout,_urate.50.as

Please revise the manuscript to rephrase the duplicated text, cite your sources, and provide details as to how the current manuscript advances on previous work. Please note that further consideration is dependent on the submission of a manuscript that addresses these concerns about the overlap in text with published work.

Response: We reviewed 5 linked sources, revised duplicate text, and supplemented and updated references. Here we need to explain the following. Regarding the issue of text overlap with published articles, we have done our best to reword the repeated text in the full text, and also tried our best to resolve the text overlap including but not limited to the Methods section to avoid the possibility of plagiarism, and marked the source of citation . However, because the systematic reviews and meta-analyses are based on the PRISMA series of standards, it is inevitable that there will be duplications in the manuscript. Finally, we very much hope that through our efforts to pass this review. 

Special thanks to you for your good comments. We tried our best to improve the manuscript and made some changes in the manuscript. These changes will not influence the content and framework of the paper.

We appreciate for warm work of editors and reviewers earnestly, and hope that the correction will meet with approval. Once again, thank you very much for your comments and suggestions.

Dear Reviewer 1:

Thank you for your comments concerning our manuscript. Those comments are all valuable and very helpful for revising and improving our paper, as well as the important guiding significance to our researches. We have done the full english language editing. Specific modifications can be quickly found based on the line number in new "Manuscript". For the full text of the revision changes, see the "Revised Manuscript with Track Changes" file. We hope that the revised new manuscript will be approved.

The authors described a protocol for systematic review and meta-analysis for

IBD and the risk of idiopathic pulmonary fibrosis. I suggest, the authors consider the following points.

1) The introduction section (paragraphs two and three) has some grammatical problems. Language editing is required.

Response: The whole text has been significantly revised by a native English-speaking language editor with a scientific writing certificate, and the english wording, grammar problems, language articulation and other issues have been solved.

2) Trim and fill analysis would be helpful in case of existing publication bias.

Response: We are sorry for our negligence of it, and this suggestion has been adopted and supplemented in the manuscript (line 189-190).

Special thanks to you for your good comments. We tried our best to improve the manuscript and made some changes in the manuscript. These changes will not influence the content and framework of the paper.

We appreciate for warm work of editors and reviewers earnestly, and hope that the correction will meet with approval. Once again, thank you very much for your comments and suggestions.

Dear Reviewer 2:

Thank you for your comments concerning our manuscript. Those comments are all valuable and very helpful for revising and improving our paper, as well as the important guiding significance to our researches. We have done the full english language editing. Specific modifications can be quickly found based on the line number in new "Manuscript". For the full text of the revision changes, see the "Revised Manuscript with Track Changes" file. We hope that the revised new manuscript will be approved.

Inflammatory bowel disease and risk of idiopathic pulmonary fibrosis, a protocol for systematic review and meta-analysis

This article presents the study protocol of a systematic review and meta-analysis aims to find the association between inflammatory bowel disease (IBD) and idiopathic pulmonary fibrosis (IPF) risk.

The subject is novel and interesting, but the authors must correct and explain these points:

General comment

1. The manuscript text suffers from fragmentary sentences and poor cohesion. The quality of scientific writing needs attention. Please consider a major revision of the manuscript by a native English speaker with a demonstrated history of scientific writing.

Response: The whole text has been significantly revised by a native English-speaking language editor with a scientific writing certificate, and the english wording, grammar problems, language articulation and other issues have already been solved.

Abstract

1. Introduction: The rationale of the study is not well presented. Please revise.

Response: The theoretical basis of the introduction has been re-elaborated (line 21-27).

2. Methods: Please consider using passive voice when presenting methods (line 28-43).

Response: The passive voice is already used in the method.

3. Results: Need to be re-written.

Response: The results have been revised. After reviewing the journal format requirements and historical documents that have been published in the journal, we revised the expected results and moved them to the discussion (line 210-212).

Strengths and limitations of this study

1. Needs to be re-written.

Response: The strengths and limitations of this study have been revised. After reviewing the journal format requirements and the historical documents that have been published in the journal, we revised the advantages and limitations and moved them to the discussion (line 230-235).

Methods

1. Please include relevant references, where necessary.

Response: Because this Agreement was developed primarily in accordance with PRISMA-P, the method part was also produced in accordance with PRISMA-P, so the main reference is PRISMA-P, in addition to The PRISMA 2020 statement. Therefore, the necessary reference materials for the method part have been attached to the text and have been updated with some updates (line 88,90,130,145,151).

Discussion

1. Please review the discussion, check its storyline, and improve its coherency. It is not easy to follow in its current form.

Response: The discussion has been reviewed and the storyline has been revised to improve coherence to some extent (line 209-240).

2. Please include relevant references, where necessary.

Response: References have been supplemented and updated in the discussion (line 216-226).

Special thanks to you for your good comments. We tried our best to improve the manuscript and made some changes in the manuscript. These changes will not influence the content and framework of the paper.

We appreciate for warm work of editors and reviewers earnestly, and hope that the correction will meet with approval. Once again, thank you very much for your comments and suggestions.

Dear Reviewer 3:

Thank you for your comments concerning our manuscript. Those comments are all valuable and very helpful for revising and improving our paper, as well as the important guiding significance to our researches. We have done the full english language editing. Specific modifications can be quickly found based on the line number in new "Manuscript". For the full text of the revision changes, see the "Revised Manuscript with Track Changes" file. We hope that the revised new manuscript will be approved.

Thanks for sending me this interesting SRMA protocol. I think the quality of this paper is suitable for publication. Here are some suggestions:

1. Abbreviations should be in complete form at their first appearance.

Response: Abbreviations have been added to the full name at first occurrence.

2. The English wording and the narrative synthesis should be improved.

Response: The whole text has been significantly revised by a native English-speaking language editor with a scientific writing certificate, and the english wording, grammar problems, language articulation and other issues have been solved. The comprehensive narrative during the discussion has been revised.

Special thanks to you for your good comments. We tried our best to improve the manuscript and made some changes in the manuscript. These changes will not influence the content and framework of the paper.

We appreciate for warm work of editors and reviewers earnestly, and hope that the correction will meet with approval. Once again, thank you very much for your comments and suggestions.

---

## [Decision Letter · Decision Letter 1]

8 Jun 2022

Inflammatory bowel disease and risk of idiopathic pulmonary fibrosis: a protocol for systematic review and meta-analysis

PONE-D-22-02613R1

Dear Dr. Li,

We’re pleased to inform you that your manuscript has been judged scientifically suitable for publication and will be formally accepted for publication once it meets all outstanding technical requirements.

Kind regards,

Negar Rezaei, M.D., Ph.D.,

Academic Editor

PLOS ONE

Reviewers' comments:

Reviewer's Responses to Questions

**Comments to the Author**

1. Does the manuscript provide a valid rationale for the proposed study, with clearly identified and justified research questions?

Reviewer #2: Yes

Reviewer #3: Yes

2. Is the protocol technically sound and planned in a manner that will lead to a meaningful outcome and allow testing the stated hypotheses?

Reviewer #2: Yes

Reviewer #3: Yes

3. Is the methodology feasible and described in sufficient detail to allow the work to be replicable?

Reviewer #2: Yes

Reviewer #3: Yes

4. Have the authors described where all data underlying the findings will be made available when the study is complete?

Reviewer #2: No

Reviewer #3: Yes

5. Is the manuscript presented in an intelligible fashion and written in standard English?

Reviewer #2: Yes

Reviewer #3: Yes

6. Review Comments to the Author

You may also provide optional suggestions and comments to authors that they might find helpful in planning their study.

Reviewer #2: I would like to thank the author for the extensive revision of the manuscript. I think the quality of the manuscript has dramatically improved.

Reviewer #3: thanks again for sending me this revision paper. I read this again with great interest and believe that all my comments are addressed pretty well and have no further concerns.

7. PLOS authors have the option to publish the peer review history of their article (what does this mean?). If published, this will include your full peer review and any attached files.

Reviewer #2: **Yes: **Mohsen Abbasi-Kangevari

Reviewer #3: **Yes: **Esmaeil Mohammad, MD MPH

---

## [Editor Report · Acceptance letter]

17 Jun 2022

PONE-D-22-02613R1 

Inflammatory bowel disease and risk of idiopathic pulmonary fibrosis: a protocol for systematic review and meta-analysis 

Dear Dr. Li:

I'm pleased to inform you that your manuscript has been deemed suitable for publication in PLOS ONE. Congratulations! Your manuscript is now with our production department. 

Kind regards, 

on behalf of

Dr. Negar Rezaei 

Academic Editor

PLOS ONE